# Vaginal Cuff Dehiscence and a Guideline to Determine Treatment Strategy

**DOI:** 10.3390/jpm13060890

**Published:** 2023-05-24

**Authors:** Kyung Jin Eoh, Young Joo Lee, Eun Ji Nam, Hye In Jung, Young Tae Kim

**Affiliations:** 1Department of Obstetrics and Gynecology, Yongin Severance Hospital, Yonsei University College of Medicine, Yongin 16995, Republic of Korea; kjeoh2030@yuhs.ac; 2Department of Obstetrics and Gynecology, Institute of Women’s Medical Life Science, Yonsei Cancer Center, Severance Hospital, Yonsei University College of Medicine, Seoul 03722, Republic of Korea; leeyj_0907@yuhs.ac (Y.J.L.); nahmej6@yuhs.ac (E.J.N.);

**Keywords:** hysterectomy, minimally invasive surgical procedure, surgical wound dehiscence, complication

## Abstract

In this retrospective study, our aim was to investigate a novel treatment strategy guideline for vaginal cuff dehiscence after hysterectomy based on the mode of operation and time of occurrence in patients who underwent hysterectomy at Severance Hospital between July 2013 and February 2019. We analyzed the characteristics of 53 cases of vaginal cuff dehiscence according to the mode of hysterectomy and time of occurrence. Out of a total of 6530 hysterectomy cases, 53 were identified as vaginal cuff dehiscence (0.81%; 95% confidence interval: 0.4–1.6%). The incidence of dehiscence after minimally invasive hysterectomy was significantly higher in patients with benign diseases, while malignant disease was associated with a higher risk of dehiscence after transabdominal hysterectomy (*p* = 0.011). The time of occurrence varied significantly based on menopausal status, with dehiscence occurring relatively earlier in pre-menopausal women compared to post-menopausal women (93.1% vs. 33.3%, respectively; *p* = 0.031). Surgical repair was more frequently required in cases of late-onset vaginal cuff dehiscence (≥8 weeks) compared to those with early-onset dehiscence (95.8% vs. 51.7%, respectively; *p* < 0.001). Patient-specific factors, such as age, menopausal status, and cause of operation, may influence the timing and severity of vaginal cuff dehiscence and evisceration. Therefore, a guideline may be indicated for the treatment of potentially emergent complications after hysterectomy.

## 1. Introduction

Hysterectomy, which is the second most common gynecological surgery worldwide after cesarean section, can have various complications depending on the surgical method and technique. These complications include hemorrhage, urinary tract and gastrointestinal injuries, post-operative fever, and vaginal cuff prolapse [1,2,3]. Vaginal cuff dehiscence, which is a rare but serious complication, occurs with an estimated incidence ranging from 0.14% to 4.1%, regardless of the hysterectomy method used [4,5,6,7,8,9]. Mild cases may be managed conservatively; however, vaginal cuff dehiscence should be treated as a surgical emergency due to the potential for partial or total evisceration, bowel strangulation, sepsis, and acute mesenteric ischemia. Vaginal evisceration, i.e., the complete protrusion of abdominal contents through a disrupted vaginal cuff, is a rare but life-threatening emergency requiring immediate management, including patient assessment, stabilization, and surgical repair of the abdominal wall defect [10,11,12,13].

Research suggests that a minimally invasive hysterectomy (MIH), including laparoscopic and robot-assisted procedures, is associated with higher rates of vaginal cuff dehiscence compared to transvaginal hysterectomy (TVH) or transabdominal hysterectomy (TAH) [4]. Past systematic reviews consistently reported 5 to 10 times higher incidence of vaginal cuff dehiscence in MIH compared to TVH or TAH [4,14]. However, some studies showed lower incidence rates following total laparoscopic hysterectomy compared to open hysterectomy [15,16].

The time of occurrence for vaginal cuff dehiscence varies based on the surgical method. Uccella et al. reported median times of 1 month (range 1–12 months) for MIH, 5 months (range 2–48 months) for TAH, and 24 months (range 1–62 months) for TVH. Dehiscence occurred earlier after laparoscopic hysterectomy compared to abdominal hysterectomy [17]. The median interval between the initial hysterectomy and the onset of dehiscence was 11 weeks, with values ranging from 1 to 13 months.

In this study, we reviewed 6530 hysterectomy cases over a 6-year period at a single center to compare the incidence of vaginal cuff dehiscence based on the surgical method. We conducted a detailed analysis of 53 patients who experienced vaginal cuff dehiscence, with or without evisceration, and explored a new treatment strategy for managing vaginal cuff dehiscence after hysterectomy.

## 2. Materials and Methods

### 2.1. Patient Selection and Data Collection

We conducted a retrospective review of patient records from Severance Hospital, covering the period between July 2013 and February 2019. Electronic medical records were examined to gather clinical data on patients who had undergone open, total laparoscopic, or robotic hysterectomy for benign or malignant conditions. The collected information included pre-operative factors (age, body mass index (BMI), menopausal status, pre-operative hemoglobin level, and previous chemotherapy history) and perioperative characteristics (pathologic diagnosis, uterine weight, mode of surgery, perioperative transfusion, and length of hospitalization). Vaginal cuff dehiscence was defined as a partial or full-thickness opening of the anterior and posterior edges of the vaginal cuff. Vaginal evisceration was diagnosed during pelvic examination when a disruption of the vaginal cuff and prolapse of intra-abdominal contents through the cuff defect were observed. Open and laparoscopic hysterectomies were performed according to established procedures [18,19,20]. Outcome variables investigated included vaginal dehiscence, the time to dehiscence, and the methods of repair. Patients who underwent conversion to an open procedure or laparoscopy-assisted hysterectomy were excluded from the analysis. Vaginal and laparoscopy-assisted vaginal hysterectomies were also excluded as they did not involve a peritoneal approach for the colpotomy incision. Similarly, patients who underwent minimally invasive surgery without colpotomy were not included in the analysis. We carefully reviewed the medical records of patients with dehiscence and evisceration to ensure that only those individuals who experienced vaginal cuff dehiscence with evisceration as a complication of hysterectomy were included in the study (Figure 1).

Colpotomy of the vaginal fornices was performed in both open and laparoscopic cases using either a scalpel or a monopolar spatula set at 60 W with a pure cutting waveform. The specific techniques for cuff closure varied based on the surgeon’s preferences. For laparoscopic hysterectomy, cuff closure involved interrupted polyglactin 0 sutures secured with polyglactin 2-0. Robot-assisted hysterectomies were performed using the da Vinci Surgical System. Colpotomy in robotic surgery was carried out using monopolar scissors set at 35 W in blended mode. The vaginal cuff was then reapproximated using either 0 Vicryl sutures placed in a figure-of-eight manner, with intracorporeal knot tying, or 0 Vicryl sutures using the Endo Stitch device with extracorporeal knot tying, depending on the surgeon’s preference.

### 2.2. A Guideline of Vaginal Cuff Dehiscence and Evisceration

Early diagnosis and prompt management of vaginal cuff dehiscence or evisceration are crucial in preventing further complications and promoting healing. This issue occurs when the surgical incision on the vagina reopens, which can lead to various complications, such as infection, hemorrhage, and injury to the bowel or bladder. However, there is currently no standard recommendation for the ideal method of surgical repair following vaginal cuff dehiscence or evisceration. The available scientific evidence on the approach (vaginal, abdominal, or laparoscopic) to repairing vaginal cuff dehiscence does not indicate a preferred method. Several factors influence the choice of management, including the patient’s vital stability, suspicion of intra-abdominal organ damage, presence of bowel evisceration, ability to evaluate bowel health, ability to adequately visualize and reapproximate vaginal tissue, surgeon availability, and the potential need for additional intensive care. Since no single method demonstrates superiority, the experienced surgeon decides on the closure technique based on their judgment of how to achieve optimal tissue approximation, strength of repair, and ability to identify additional issues.

### 2.3. Statistical Analysis

SPSS version 26.0 (IBM Corp., Armonk, NY, USA) was used for the statistical analysis. Normal distribution was evaluated using the Shapiro–Wilk test. Continuous variables were compared using Student’s *t*-test or the non-parametric Mann–Whitney test when a normal distribution could not be assumed. Fisher’s exact test was used to compare the proportions. Two-tailed statistical significance was set at 0.05.

### 2.4. Ethics

This cohort study was approved by the Institutional Review Board (IRB) of Yonsei University of Korea (approval code 4-2023-0286) The IRB waived the requirement for written informed consent. This study was conducted in accordance with the principles of the Declaration of Helsinki.

## 3. Results

A total of 6530 hysterectomy cases were identified. Of these, 3893 patients underwent MIH, including 434 robot-assisted hysterectomies (59.6%), and 2637 patients underwent TAH (40.4%). Overall, 53 cases of vaginal cuff dehiscence occurred (0.81%, 95% confidence interval (CI): 0.4–1.6%), with 21 cases observed among patients who underwent TAH (0.46%, 95% CI: 0.1–1.6%) and 41 cases observed among patients who underwent MIH (1.05%, 95% CI: 0.0–1.6%).

Based on the medical records analyzed, 53 of the 6530 patients who underwent various modes of hysterectomy experienced vaginal cuff dehiscence with or without evisceration. The patients’ clinical characteristics are presented in Table 1.

A significant difference was observed regarding the indications for hysterectomy when the clinical characteristics of patients who underwent TAH and MIH were compared. The incidence of vaginal cuff dehiscence after MIH in patients with benign diseases was significantly higher than that in patients with malignant diseases (73.2% vs. 26.8%, respectively). In contrast, malignant diseases were associated with a higher risk of vaginal dehiscence after TAH than benign diseases (66.7% vs. 33.3%, respectively; *p* = 0.011). Among patients treated with chemotherapy, the incidence was significantly higher in the TAH group than in the MIH group (*p* = 0.036). In patients who underwent MIH, the uterus tended to be heavier; however, the difference was insignificant (*p* = 0.457). There were no significant differences with respect to age, menopausal status, BMI, hemoglobin level, or parity.

The mean time intervals between the initial hysterectomy and the onset of dehiscence were 93.3 days after TAH and 67.2 days after MIH. Vaginal cuff dehiscence occurred earlier in the MIH group than in the TAH group; however, the difference was insignificant in this study. Analysis of patients’ perioperative characteristics revealed that evisceration, treatment method, post-cuff repair complications, transfusion, and length of hospitalization did not differ between the MIH and TAH groups. Six patients presented with the evisceration of various protruding organs. No cases of vaginal cuff evisceration were observed in the patients who underwent robot-assisted hysterectomy (Table 2).

Table 3 presents the clinical characteristics of the patients who experienced cuff dehiscence with respect to the time to occurrence: early (<8 weeks) and late (≥8 weeks). Late occurrence was significantly more common (*p* = 0.031) in post-menopausal women, with 27 cases occurring before 8 weeks in pre-menopausal women (93.1%) and 8 cases occurring after 8 weeks in post-menopausal women (33.3%). The management methods for vaginal cuff dehiscence were divided into four categories: laparoscopic, open, vaginal, and expectant (Table 3).

In this study, the repair method significantly varied based on the time to occurrence (*p* < 0.001), with surgical repair being performed more frequently in patients with late occurrence of cuff dehiscence (*p* < 0.001). A case of vaginal cuff dehiscence with evisceration following MIH is illustrated in Figure 1. The patient underwent laparoscopic surgery to repair the ruptured vaginal cuff.

In patients who underwent hysterectomy for benign causes, age was positively correlated with time to occurrence (correlation 0.51, *p* < 0.05), whereas BMI (correlation −0.22, *p* = 0.22) was not positively correlated with time to occurrence (Figure 2).

## 4. Discussion

Our study demonstrated that vaginal cuff dehiscence is a relatively rare event, occurring in approximately 0.81% of patients who underwent hysterectomy through different modes at our tertiary academic medical center. To the best of our knowledge, this is the largest cohort study investigating the incidence of vaginal cuff dehiscence and its risk factors among patients undergoing different modes of hysterectomy at a single tertiary referral institution. This occurrence rate is similar to previously reported rates of 0.14–4.1% [4,5,21]. Previous reports also suggested that the incidence of vaginal cuff dehiscence after robotic hysterectomy is comparable to that after laparoscopic hysterectomy [22], resulting in a higher incidence following robotic surgery than open surgery. Our single-center study revealed no cases of vaginal cuff evisceration in patients who underwent robotic hysterectomy. Similarly, Kashani et al. reported that the incidence of vaginal cuff dehiscence after robotic hysterectomy was comparable to that after transabdominal and vaginal hysterectomies [16].

Potential risk factors for cuff dehiscence include age, menopausal status, BMI, cuff closure approach, malignancy, and sexual intercourse before wound healing [21,23]. In our cohort, malignant diseases were associated with a higher risk of vaginal dehiscence after transabdominal hysterectomy than benign diseases. Ceccaroni et al. reported malignancy as a significant independent risk factor for vaginal cuff dehiscence with evisceration, with an incidence of 0.8% after total hysterectomy for malignant indications compared to 0.2% after total hysterectomy for pelvic prolapse [24]. Additionally, vaginal atrophy, factors associated with poor wound healing (such as obesity, malignancy, radiation, chronic steroid use, and previous vaginoplasty), post-operative infection or hematoma, and maneuvers that increase abdominal pressure (including chronic cough, constipation, and Valsalva maneuver) were also implicated.

Patients requiring total hysterectomy for malignant disease are at higher risk of complications, including vaginal cuff dehiscence, due to a combination of malnutrition and multiple comorbidities. In a case series published by Drudi et al., patients who received post-operative adjuvant therapy had a higher incidence of vaginal cuff dehiscence (3%) compared to those who did not receive adjuvant treatment (0.4%). Furthermore, the use of adjuvant chemotherapy was shown to delay wound healing and resulted in weakening of the vaginal apex, which is a known confounding factor in several studies of vaginal cuff dehiscence [6].

In locally advanced cervical cancer, primary chemoradiation therapy had a high incidence of residual disease (up to 50%), increasing the risk of local recurrence. Completion surgery in these patients also carries an additional risk of vaginal cuff dehiscence due to poor tissue quality. Therefore, careful management and monitoring are necessary. Prompt diagnosis, surgical repair, and regular follow-up appointments are essential to prevent complications and ensure proper healing. Nutritional support and monitoring of comorbidities are important to reduce further risks. Careful surgical technique and wound closure methods should be employed to minimize vaginal cuff dehiscence. Patient education is crucial for recognizing and reporting signs of dehiscence. The use of adjuvant therapy should be carefully considered, weighing the benefits against the risks, including vaginal cuff dehiscence.

Similar to previous studies [16], vaginal cuff dehiscence occurred earlier in patients in the MIH group than in those in the TAH group; however, this difference was not significant in this study. Menopausal status affects the timing of dehiscence occurrence. Young pre-menopausal women may be more sexually active, triggering dehiscence often associated with sexual intercourse before complete healing. In our series, all but two pre-menopausal women reported intercourse as a triggering event, with an early onset and no major complications. Delayed wound healing due to vaginal atrophy may result in late-onset dehiscence in post-menopausal women. These findings help clinicians counsel patients post-operatively. Late-onset dehiscence often requires surgical management, warranting further studies on interventions to reduce its rate.

Vaginal cuff dehiscence can be repaired vaginally, abdominally, laparoscopically, or through a combined approach [25]. In our series, 39% of dehiscences were repaired vaginally (n = 21), 30% were repaired laparoscopically (n = 16), and 4% were repaired abdominally. The surgical repair techniques in the laparoscopic approach consisted of knot tying with interrupted absorbable sutures of 0 polyglactin (Coated VICRYL^®^ Plus, ETHICON, Somerville, NJ, United States) placed approximately 5 mm apart. Nine closures were performed with monofilament suture during a transition to the barbed suture technique, according to the surgeon’s preference. There were no instances of post-operative infection and re-dehiscence in the entire surgical repair approach group. The vaginal approach seemed minimally invasive but precluded observation of the entire abdominal cavity and irrigation of probable abscesses. The use of antibacterial monofilament absorbable suture facilitates vaginal stump closure in laparoscopic hysterectomy without increasing complications, such as cuff dehiscence, especially in less experienced operators. On the other hand, laparoscopic repair reduces the chances of operative injury to eviscerated bowel loops, which can occur during vaginal repair. It allows thorough inspection of the abdominal cavity and entire bowels. Early laparoscopic repair prevents complications, such as ascending peritonitis, bowel ischemic injury, and recurrence of cuff dehiscence [26,27].

There is no standard recommendation for the ideal surgical repair method for vaginal cuff dehiscence or evisceration. The vaginal approach was the most commonly used surgical repair method in our study. Similarly, Cronin et al. reported that 51% of dehiscences were repaired vaginally, 32% abdominally, 2% laparoscopically, and 10% using a combined (abdominal, vaginal, or laparoscopic and vaginal) approach; a further 5% were allowed to heal through expectant management [20]. The clinical evidence available on the approach (vaginal, abdominal, and laparoscopic) for the surgical repair of vaginal cuff dehiscence does not suggest that one approach is preferred over the others. Several factors influence the choice of surgical repair, including the vital stability of the patient, the surgeon’s experience, the presence of bowel evisceration, the ability to evaluate the bowel for ischemia or strangulation, and the ability to perform additional necessary procedures. Unexpected vaginal evisceration can be dreadful for the patient and unforgettable for the gynecologist. Thus, we suggest using a guideline to treat this potentially emergent complication after hysterectomy. This recommendation was extracted directly from relevant surgical events to support clinical decision-making related to conservative or immediate surgical treatment of patients with vaginal cuff dehiscence and evisceration (Figure 3).

Our study has several limitations. Firstly, it is a retrospective study, specifically focusing on the occurrence of cuff dehiscence after hysterectomy. We analyzed cases of vaginal cuff dehiscence within a specific analysis period based on certain criteria, which may have resulted in the exclusion of some instances of dehiscence. We found that the average time of occurrence was 80 days; however, we observed cases occurring as late as 2000 days in patients who underwent adjuvant radiotherapy, which were not included in our analysis. Secondly, the study was conducted in a single center with a limited number of patients. The low incidence of vaginal cuff dehiscence in our sample could introduce bias. Therefore, future research should involve larger multicenter studies to obtain more comprehensive results.

## 5. Conclusions

In conclusion, our findings were consistent with the idea that patient-specific factors, including age, menopausal status and cause of operation, may influence the timing and severity of vaginal cuff dehiscence and evisceration. Therefore, a guideline may be indicated to treat these potentially emergent complications after hysterectomy.

## Figures and Tables

**Figure 1 jpm-13-00890-f001:**
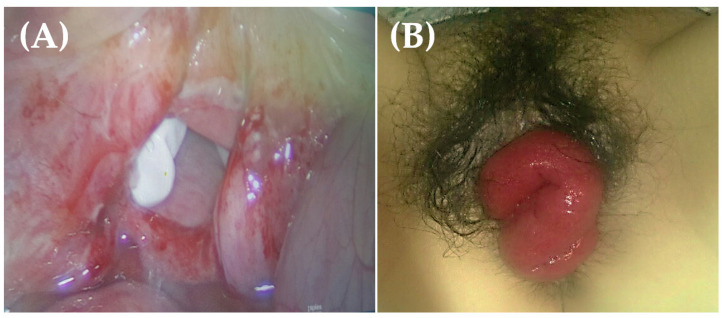
(**A**) Vaginal cuff reveals complete dehiscence, and a white surgical glove is visualized on vaginal vault. (**B**) Small bowel protruding through completely ruptured vagina. Vaginal cuff dehiscence and evisceration were corrected laparoscopically without complications.

**Figure 2 jpm-13-00890-f002:**
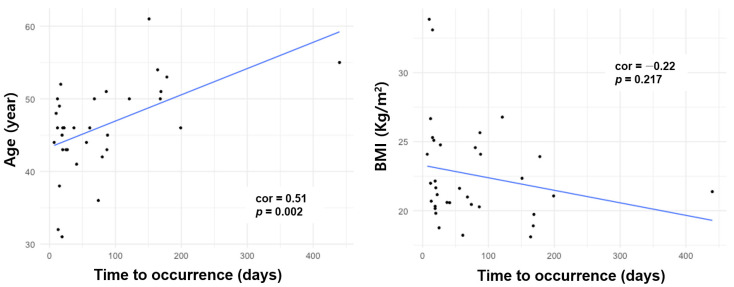
Scatter plots of age and BMI with respect to time to cuff surgery for patients undergoing cuff repair surgery after hysterectomy for benign disease.

**Figure 3 jpm-13-00890-f003:**
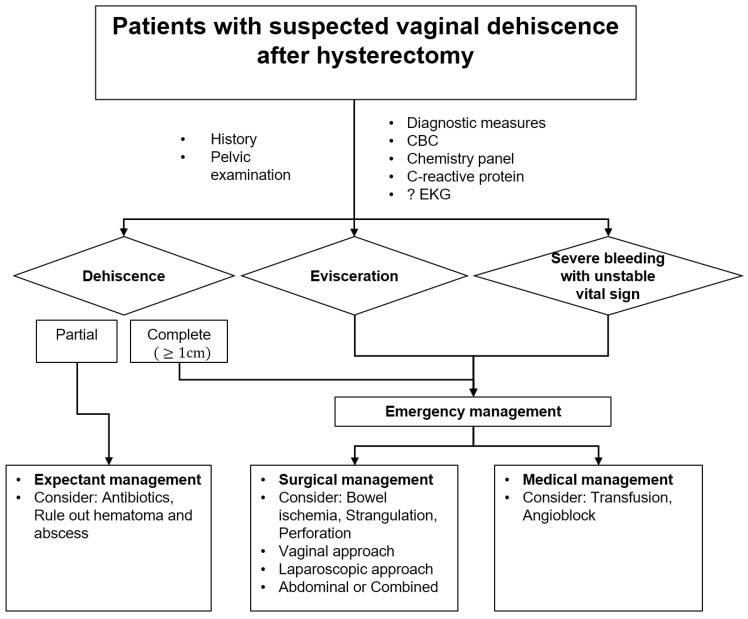
Novel guideline for treatment strategy for vaginal cuff dehiscence and evisceration after hysterectomy.

**Table 1 jpm-13-00890-t001:** Baseline cohort characteristics before vaginal cuff dehiscence.

	Overall	TAH	MIH	
Variables	(n = 53)	(n = 12)	(n = 41)	*p* ^(c)^
Age (years) ^(a)^	46.6 ± 7.2	47.0 ± 7.6	46.5 ± 7.2	0.847
Menopause ^(b)^				
Pre	43 (81.1)	10 (83.3)	33 (80.5)	0.825
Post	10 (18.9)	2 (16.7)	8 (19.5)	
BMI (kg/m^2^) ^(a)^	22.5 ± 4.0	22.4 ± 4.3	22.5 ± 3.9	0.959
Diagnosis ^(b)^				
Benign	34 (64.2)	4 (33.3)	30 (73.2)	0.011
Malignancy	19 (35.8)	8 (66.7)	11 (26.8)	
Parity ^(a)^	1.8 ± 0.9	2.1 ± 1.3	1.7 ± 0.8	0.254
Initial Hb (g/dL) ^(a)^	11.8 ± 1.8	11.7 ± 1.9	11.8 ± 1.8	0.867
Uterine weight (g) ^(a)^	243.6 ± 497.3	144.7 ± 94.0	273.8 ± 564.4	0.457
Chemotherapy ^(b)^				
No	48 (90.6)	9 (75.0)	39 (95.1)	0.036
Yes	5 (9.4)	3 (25.0)	2 (4.9)	

Values are presented as mean ± standard deviation or number (%). MIH, minimally invasive hysterectomy; TAH, total abdominal hysterectomy; BMI, body mass index; Hb, hemoglobin. ^(a)^ *p*-value from analysis of student *t*-test or Mann–Whitney test; ^(b)^ *p*-value from χ^2^ test or Fisher’s exact test. ^(c)^ *p*-value was obtained through comparing only TAH and MIH group.

**Table 2 jpm-13-00890-t002:** Perioperative characteristics of patients with vaginal cuff repair.

	Overall	TAH	MIH	
Variables	(n = 53)	(n = 12)	(n = 41)	*p*
Time to occurrence (days) ^(a)(b)^	80.2 ± 73.2	93.3 ± 123.2	67.2 ± 60.8	0.671
Interval to dehiscence				
<8 weeks (early)	29 (54.7)	8 (66.7)	21 (51.2)	0.344
≥8 weeks (late)	24 (45.3)	4 (33.3)	20 (48.8)	
Evisceration ^(b)^				0.425
No	50 (94.3)	11 (83.3)	38 (90.2)	
Yes	3 (5.7)	2 (16.7)	4 (9.8)	
Method of treatment ^(b)^				0.074
Laparoscopic	16 (30.2)	1 (8.3)	15 (36.5)	
Abdominal	2 (3.8)	0 (0)	2 (4.9)	
Vaginal	21 (39.6)	9 (75.0)	12 (29.3)	
Expectant	14 (26.4)	2 (16.7)	12 (29.3)	
Post-cuff repair complication ^(b)^				0.260
No	49 (92.5)	12 (100)	37 (90.2)	
Yes	4 (7.5)	0 (0)	4 (9.8)	
Transfusion ^(b)^				0.585
No	52 (98.1)	12 (100)	40 (97.6)	
Yes	1 (1.9)	0 (0)	1 (2.4)	
Hospitalized days (days) ^(a)^	5.8 ± 2.58	5.3 ± 1.8	6.3 ± 4.9	0.730

Values are presented as mean ± standard deviation or number (%). MIH, minimally invasive hysterectomy; TAH, total abdominal hysterectomy. ^(a)^ *p*-value from analysis of student *t*-test or Mann–Whitney test; ^(b)^ *p*-value from χ^2^ test or Fisher’s exact test.

**Table 3 jpm-13-00890-t003:** Comparison between early and late occurrence in patients with vaginal cuff dehiscence.

Variables	Early Occurrence < 8 Weeks	Late Occurrence ≥ 8 Weeks	*p*
Age (years) ^(a)^	45.3 ± 1.3	48.3 ± 1.5	0.130
Menopause ^(b)^			
Pre	27 (93.1)	16 (66.7)	0.031
Post	2 (6.9)	8 (33.3)	
BMI (kg/m^2^) ^(a)^	23.2 ± 0.8	21.6 ± 0.6	0.123
Uterine weight (g) ^(a)^	322.3 ± 645.2	137.3 ± 91.2	0.211
Diagnosis ^(b)^			
Benign	18 (62.1)	16 (66.7)	0.728
Malignancy	11 (37.9)	8 (33.3)	
Chemotherapy ^(b)^			
No	26 (89.7)	22 (21.7)	1.000
Yes	3 (10.3)	2 (8.3)	
Mode of hysterectomy ^(b)^			
MIH	21 (72.4)	20 (83.3)	0.344
TAH	8 (27.6)	4 (16.7)	
Surgical repair ^(b)(c)^			
Yes	16 (51.7)	23 (95.8)	0.000
No	13 (48.3)	1 (4.2)	
Method of treatment ^(b)^			
Laparoscopic	3 (11.1)	13 (54.2)	0.000
Abdominal	1 (3.4)	1 (4.2)	
Vaginal	12 (41.4)	9 (37.5)	
Expectant	13 (44.8)	1 (4.2)	
Hospitalized days after dehiscence (days) ^(a)^	7.1 ± 5.2	4.9 ± 2.6	0.149

Values are presented as mean ± standard deviation or number (%). BMI, body mass index; MIS, minimally invasive surgery. ^(a)^ *p*-value from analysis of student *t*-test or Mann–Whitney test; ^(b)^ *p*-value from χ^2^ test or Fisher’s exact test. ^(c)^ Surgical repair includes MIH, open, and vaginal approaches for repair.

## Data Availability

Due to the nature of this retrospective study, participants in this study did not agree for their data to be shared publicly; thus, supporting data are not available.

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
