# Peer review of "Vaginal Cuff Dehiscence and a Guideline to Determine Treatment Strategy"

_jpm, 2023, doi:10.3390/jpm13060890_

Round 1
Reviewer 1 Report
Dear colleagues
Thank you for your very good paper. It is well structured, easy to read and understand
Unfortunately, the paper is not publishable in this format because there is a very significant discrepancy between the title and the content. You read the title and you expect to read about a novel algorithm but you come across a retrospective study about the incident of vaginal dehiscence post hysterectomy. The algorithm appears only as a table at the end of the discussion. My recommendation is to change the title or the whole structure of the paper. I would love to read about the novel algorithm, how your data helped you to develop this algorithm, how this new tool is tested and how it changed patients' outcomes. On the other hand, I appreciate that this is a massive task and I accept a paper with a title like " Vaginal cuff dehiscence and an algorithm to determine treatment strategy". In that case, the algorithm and the rational behind every step should be very clearly explained in the methodology.
Another comment is regarding the discussion of the paper. You present numbers but there is lack of details regarding the reason behind these numbers. For example, why you don't have any dehiscence after robotic cases?, why the dehiscence is higher in malignancies?, it is only because of chemo or something else?
Also, the discussion needs a separate paragraph for the surgical repair method, not only the approach. And, of course, the outcome of the repair ( successful or not, peri-op complications, dehiscence post repair etc).
minor changes required
Author Response
Thank you for your very good paper. It is well structured, easy to read and understand
Unfortunately, the paper is not publishable in this format because there is a very significant discrepancy between the title and the content. You read the title and you expect to read about a novel algorithm but you come across a retrospective study about the incident of vaginal dehiscence post hysterectomy. The algorithm appears only as a table at the end of the discussion. My recommendation is to change the title or the whole structure of the paper.
→ Thank you very much for your valuable comments.
We changed the title of paper (vaginal cuff dehiscence and an algorithm to determine treatment strategy).
I would love to read about the novel algorithm, how your data helped you to develop this algorithm, how this new tool is tested and how it changed patients' outcomes. On the other hand, I appreciate that this is a massive task and I accept a paper with a title like " Vaginal cuff dehiscence and an algorithm to determine treatment strategy". In that case, the algorithm and the rationale behind every step should be very clearly explained in the methodology.
→ Thank you very much for your valuable comments.
We elaborated on the algorithmic management of vaginal cuff dehiscence and evisceration in the metholodgy;
2.2. Algorithmic management of vaginal cuff dehiscence and evisceration
Early diagnosis and prompt management of vaginal cuff dehiscence or evisceration are essential to prevent further complications and promote healing. It occurs when the surgical incision on the vagina reopens, which can lead to a range of complications such as infection, hemorrhage, and bowel or bladder injury. However, there is no standard recommendation about the ideal method of surgical repair after vaginal cuff dehiscence or evisceration. The scientific evidence that is available on the approach (vaginal, abdominal, laparoscopic) to the repair of a vaginal cuff dehiscence does not suggest that one approach is preferred over the others. Several factors affect the choice of management: the vital stability of the patient, level of suspicion for damage to intraabdominal organs, whether a bowel evisceration is present, the ability to evaluate the bowel for ischemia or damage, the ability to visualize and reapproximate vaginal mucosa adequately, surgeon availability, and the ability to perform additional intensive care. Because no single method is superior to another, the method of closure is decided by the experienced surgeon on the basis of which management he or she thinks will allow the best tissue approximation, strength of repair, and ability to assess for additional problems
Another comment is regarding the discussion of the paper.
You present numbers but there is lack of details regarding the reason behind these numbers. For example, why you don't have any dehiscence after robotic cases?,
→ Thank you for your comments.
We described that no cases of vaginal cuff evisceration were observed in the patients who underwent robot-assisted hysterectomy. However, we experienced 3 cases with vaginal cuff dehiscence. Two of them underwent laparoscopic repair and one of them had expectant management.
why the dehiscence is higher in malignancies?, it is only because of chemo or something else?
→ We inserted these explanations in the discussion section:
Patientnts who require total hysterectomy for malignant disease are at a higher risk of complications, including vaginal cuff dehiscence, due to a combination of malnutrition and multiple comorbidities. In a case series published by Drudi et al., patients who received postoperative adjuvant therapy had a higher incidence of vaginal cuff dehiscence (3%) compared to those who did not receive adjuvant treatment (0.4%). Furthermore, the use of adjuvant chemotherapy was shown to delay wound healing and result in weakening of the vaginal apex, which is a known confounding factor in several studies of vaginal cuff dehiscence.
In cases of locally advanced cervical cancer, primary chemoradiation therapy has been shown to have a high incidence of residual disease (up to 50%), which increases the risk of local recurrence. Completion surgery in these patients also carries an additional risk of vaginal cuff dehiscence due to the poor tissue quality of the vaginal cuff. Therefore, careful consideration and close monitoring are required when managing patients with locally advanced cervical cancer who undergo hysterectomy and adjuvant therapy. The management of vaginal cuff dehiscence in patients with malignant disease requires more intensive care due to the increased risk of complications. Prompt diagnosis and surgical repair are essential to prevent further complications and proper healing. Patients may require regular follow-up appointments to assess the healing of the wound and monitor for signs of infection or other complications. In addition, patients may need nutritional support and close monitoring of their comorbidities to reduce the risk of further complications. To reduce the risk of vaginal cuff dehiscence in patients with malignant disease, careful surgical technique and appropriate wound closure methods should be employed. Patient education is also crucial in ensuring that patients are aware of the signs and symptoms of dehiscence and the importance of reporting any concerns to their healthcare provider. Furthermore, the use of adjuvant therapy should be considered carefully, weighing the potential benefits against the risk of complications, including vaginal cuff dehiscence.
Patients with malignant disease who undergo hysterectomy and adjuvant therapy are at a higher risk of vaginal cuff dehiscence due to malnutrition and multiple comorbidities. The management of vaginal cuff dehiscence in these patients requires prompt diagnosis, surgical repair, and close monitoring, with nutritional support and careful consideration of adjuvant therapy to reduce the risk of complications.
Drudi, L.; Press, J.Z.; Lau, S.; Gotlieb, R.; How, J.; Eniu, I.; Drummond, N.; Brin, S.; Deland, C.; Gotlieb, W.H. Vaginal vault dehiscence after robotic hysterectomy for gynecologic cancers: Search for risk factors and literature review. International journal of gynecological cancer : official journal of the International Gynecological Cancer Society 2013, 23, 943-950.
Also, the discussion needs a separate paragraph for the surgical repair method, not only the approach. And, of course, the outcome of the repair (successful or not, peri-op complications, dehiscence post repair etc).
→ We also added a separate paragraph for the surgical repair method and outcome of the repair in the discussion session;
Vaginal cuff dehiscence can be repaired vaginally, abdominally, laparoscopically, or through a combined approach. In our series, 39% of dehiscence were repaired vaginally (n=21), 30% were repaired laparoscopically (n=16), and 4% were repaired abdominally. Surgical repair techniques in laparoscopic approach consisted of knot tying with interrupted absorbable (about 5 mm apart) sutures of 0 polyglactin (Coated VICRYL® Plus, ETHICON). Nine closures were performed with monofilament suture during a transition to the barbed suture technique according to the preference of surgeon. There were no instances of postoperative infection and redehiscence in entire surgical repair approach group. The vaginal approach seemed to be minimally invasive but precludes observation of the entire abdominal cavity and irrigation of probable abscess. Use of antibacterial monofilament absorbable suture facilitates vaginal stump closure in laparoscopic hysterectomy without increasing the complications, such as cuff dehiscence, especially in less experienced operators. On the other hand, laparoscopic repair reduces the chances of operative injury to eviscerated bowel loop, which can occur during vaginal repair. It allows thorough inspection of the abdominal cavity and entire bowels. Early laparoscopic repair prevents complications such as ascending peritonitis and bowel ischemic injury, and recurrence of cuff dehiscence.
Reviewer 2 Report
The authors present a retrospective analysis on the incidence of vaginal cuff dehiscence following hysterectomy. The paper presents the possibilities of patient management for a postoperative complication that is quite rare and that may present at a significant period following the surgical procedure. Therefore, a set of guiding steps in the management of vaginal cuff dehiscence is welcomed. The text is well written and easy to read.
Some minor comments:
- for Figure 3 please add the meaning of the R/O abbreviation;
- at the end of the manuscript please add a paragraph for the conclusion section.
Author Response
The authors present a retrospective analysis on the incidence of vaginal cuff dehiscence following hysterectomy. The paper presents the possibilities of patient management for a postoperative complication that is quite rare and that may present at a significant period following the surgical procedure. Therefore, a set of guiding steps in the management of vaginal cuff dehiscence is welcomed.
The text is well written and easy to read.
Some minor comments:
- for Figure 3 please add the meaning of the R/O abbreviation;
→ R/O = rule out
We changed the abbreviation.
- at the end of the manuscript please add a paragraph for the conclusion section.
→ We added the conclusion section in the end of the manuscript.
Round 2
Reviewer 1 Report
dear authors
thank you for your paper
unfortunately, despite some significant improvements, the paper is not suitable for publication. The main problem, which is the lack of a clear algorithm, remains unsolved.
You added the 2.2 paragraph in the Methods but you still don't present an algorithm. Algorithm is a clear tool, a flowchart to help decision making. These is nothing like that in your paper. You say that " . Several factors affect the choice of management: the vital stability of the patient, level of suspicion for damage to intraabdominal organs etc"
This is not an algorythm!
Algorythm is: when the patient is stable, I ll do this, when I have damage to intra abdominal organs I ll do the other.
And all these options are better to be on a flowchart to navigate your reader,
And you conclude that "Because no single method is superior to another, the method of closure is decided by the experienced surgeon on the basis of which management he or she thinks will allow the best tissue approximation, strength of repair, and ability to assess for additional problems"
What I can understand from this statement is that there is no algorithm but the surgeons do whatever they believe is the best for every occasion!
PLEASE, present an algorithm or change the title and the abstract of the paper
In addition from lines 259-289 you talk about risk of dehiscence in patients with malignancies. The same information are mentioned 2-3 times, please write one paragraph avoiding duplications.
Finally, minor editing of english language is needed
minor editing is needed
Author Response
We thank you for the important and valuable comments on our manuscript. We have revised our manuscript according to the reviewers' comments as follows.
<Comments and Suggestions for Authors>
Unfortunately, despite some significant improvements, the work is not suitable for publication. The main problem, namely the lack of a clear algorithm, remains unsolved.
You have added paragraph 2.2 to methods, but you still do not present an algorithm. An algorithm is a clear tool, a flowchart that helps in decision making. There is nothing of this in your paper. You say that "several factors influence the choice of management: the vital stability of the patient, the degree of suspicion of damage to intra-abdominal organs etc."
This is not an algorythm!
Algorythm is: if the patient is stable I do this, if I have damage to intra-abdominal organs I do the other.
And all of these options would be better presented in a flowchart to help the reader navigate,
And you conclude, "Since no single method is superior to another, the experienced surgeon decides which method of closure he or she believes will provide the best approach to the tissue, the best strength of repair, and the best opportunity to evaluate additional problems.
What I understand from this statement is that there is no algorithm, but surgeons do what they think is best for the situation!
PLEASE present an algorithm or change the title and abstract of the paper.
- Thank you for your insightful comment. We agree with you that the term "algorithm" is not appropriate. After some thought, we have replaced the word "algorithm" or "algorithmic management" with the term "guidelines." We hope this is a more appropriate term.
Also, in lines 259-289, you talk about the risk of dehiscence in patients with malignancies. The same information is mentioned 2-3 times, please write a paragraph to avoid duplication.
- Thank you for the valuable comment. As suggested, we have made the part more concise.
Finally, a minor revision of the English language is required
- We had the English proofreading again by the native speaker. Thank you for your valuable advice.